# Understanding the Influence of Patient Factors on Accuracy and Decision-Making in a Diagnostic Accuracy Study with Multiple Raters—A Case Study from Dentistry

**DOI:** 10.3390/ijerph20031781

**Published:** 2023-01-18

**Authors:** Kirstin Vach, Nadine Schlueter, Carolina Ganss, Werner Vach

**Affiliations:** 1Institute of Medical Biometry and Statistics, Faculty of Medicine and Medical Center, University of Freiburg, Stefan-Meier-Str. 26, D-79104 Freiburg, Germany; 2Center for Dental Medicine, Department of Operative Dentistry and Periodontology, Faculty of Medicine and Medical Center, University of Freiburg, Hugstetter Straße 55, D-79106 Freiburg, Germany; 3Department of Conservative Dentistry, Periodontology and Preventive Dentistry, Hannover Medical School, Carl-Neuberg-Str. 1, D-30625 Hannover, Germany; 4Department for Operative Dentistry, Endodontics, and Pediatric Dentistry, Section Cariology of Ageing, Philipps-University Marburg, Georg-Voigt-Str. 3, D-35039 Marburg, Germany; 5Basel Academy for Quality and Research in Medicine, Steinenring 6, CH-4051 Basel, Switzerland

**Keywords:** diagnostic study, accuracy, multiple raters, meta-analysis, enamel, dentine, diagnostic difficulty, perception

## Abstract

In diagnostic accuracy studies, the test of interest is typically applied only once in each patient. This paper illustrates some possibilities that arise when diagnoses are carried out by a sufficiently large number of multiple raters. In a dental study, sixty-one examiners were asked to diagnose 49 tooth areas with different grades of tissue loss (minor, moderate, and advanced) to decide whether dentine was exposed (positive status) or not (negative status). The true status was determined by histology (reference). For each tooth, the rate of correct decisions reflecting the difficulty to diagnose this tooth and the positive rate reflecting the perception of the tooth by the raters was computed. Meta-analytical techniques were used to assess the inter-tooth variation and the influence of tooth-specific factors on difficulty or perception, respectively. A huge variation in diagnostic difficulty and perception could be observed. Advanced tissue loss made diagnoses more difficult. The background colour and tissue loss were associated with perception and may hint to cues used by the raters. The use of multiple raters in a diagnostic accuracy study allows detailed investigations which make it possible to obtain further insights into the decision-making process of the raters.

## 1. Introduction

The accuracy of a diagnostic test is not a constant value. It may vary with the rater (i.e., the subject applying the test), with the circumstances of performing the test, or from patient to patient. For example, the accuracy may increase with the experience of the rater, with the quality of an image, or with the absence of comorbidity [1,2]. If diagnoses are based on the visual inspection of images, the accuracy depends potentially on a complex interaction between the presence and absence of visual cues and rater characteristics. Scientists from the fields of pathology [3] and radiology [4], among others, have therefore investigated potential factors influencing the assessors in their work. These studies identified a wide range of factors ranging from cognitive biases to personality traits that influence the diagnostic process and the resulting accuracy [5]. Understanding the influence of factors at the rater, performance, or patient level on the diagnostic accuracy can be helpful to improve a test or its application [6,7]. Santini et al. [8] discusses how preconditions for applying the test can be defined or instructions can be adapted to handle specific challenging situations.

Most studies on diagnostic accuracy do not allow investigation of the influence of such factors in a meaningful manner. Of course, subgroups of patients can be compared with respect to the diagnostic accuracy. However, as diagnostic accuracy studies are typically powered to estimate the accuracy in the overall study population, such subgroup analyses yield typically inconclusive results due to lack of power [9].

The situation changes if a diagnostic accuracy study involves several raters and each rater applies the diagnostic test to all (or at least a substantial number of) patients. This opens the possibility for further investigations:It is possible to estimate (with sufficient precision) the diagnostic accuracy of each single rater. Consequently, it becomes possible to analyse the inter-rater variation in diagnostic accuracy. This may allow to identify rater-specific factors explaining variation in diagnostic accuracy such as experience or specialisation. or to identify single raters with a high accuracy who may serve as role models. Corresponding analyses will be presented in a separate paper.It is possible to compute for each patient the rate of correct decisions, which reflects the difficulty to diagnose this patient. Consequently, it becomes possible to analyse the inter-patient variation in diagnostic difficulty and to identify patient-related factors, which may explain this variation, e.g., the absence or presence of comorbidities. It is also possible to compute for each patient the rate of positive ratings, reflecting the perception of a patient by the raters with respect to their true status. The inter-patient variation in perception can be analysed, too, and patient-related factors explaining this variation may give a hint about the cues raters are using in diagnosing the patients. Such factors can be, for example, specific symptoms.

The potential to learn about the influence of patient-related factors on diagnostic difficulty and perception using the information provided by multiple raters is illustrated in this paper using data from a previously published study [10]. In the general context of assessing the tooth wear status, the study investigated the role of a visual examination in order to decide whether dentine is exposed in a defined tooth area or whether the enamel cover is still intact. This distinction was part of almost all index systems used at that time (2006) to classify tooth wear into degrees of severity [11]. Thereby it was assumed that the loss of substance of a tooth is higher when dentine is exposed than when it is not. In addition, exposed dentine may be considered as a risk indicator for progression, because it has a significantly lower microhardness than enamel [12] and may therefore be more prone to mechanical wear. Diagnosing exposed dentine is thus important for prevalence and incidence studies of tooth wear on the one hand but also for the individual care of patients in everyday dental practice. The study [10] is well suited for the present project in that it provides diagnostic decisions on a sufficiently sized number of teeth from a large number of examiners, collected under standardised conditions and for which a highly reliable reference standard is available through histological examination. The analysis of the original study indicated a sensitivity of 64.5% (95% CI: 59.6–69.4%) and a specificity of 88.4% (95% CI: 83.4–93.3%). These values indicate also room for improvement in the diagnostic process, which may be informed by insights into factors influencing the current decision- making.

Our basic approach to learn about the influence of patient-related factors is aiming at describing the degree of variation in diagnostic difficulty and perception across teeth and at explaining the variation by preselected tooth-specific factors. This will be based on standard statistical techniques well known from meta-analyses. However, we will also illustrate an exploratory approach to identify new factors, which are yet unmeasured. In addition to the principal investigator of the study (CG), a further domain expert (NS) participated in this study. Specifically, the domain experts phrased the expectations with respect to the preselected factors and provided the domain perspective to the exploratory approach. Hence, the main purpose of this paper is to show (using data from dentistry as an example) some possibilities that arise when many raters perform the same task in a diagnostic accuracy study.

## 2. Materials and Methods

### 2.1. Material

Our investigation is based on a study published in 2006 [10]. The study investigated the diagnostic accuracy of distinguishing visually between “dentine not exposed” (negative status) and “dentine exposed” (positive status). Forty-nine surface areas were judged by 61 raters.

The areas stem from a pool of extracted human teeth stored in saturated aqueous thymol solution. Forty-one teeth with signs of tooth wear of various aetiologies were selected, providing 49 areas. The areas to be assessed were numbered according to the order in which they were examined, from 1 to 49, and labelled with the tooth number, followed by “a” or “b” in cases of two areas. The areas took the role of units to be diagnosed instead of patients. The teeth were provided to each rater and, in addition, photographs (with a 10-fold magnification) were presented with the area to be diagnosed marked with a sticker. This way the raters were guided to judge only the area of interest. Example teeth with marked areas are shown in Figure 1.

The raters were recruited from three different settings. The group consisted of 23 scientists, 18 university dentists, and 20 dental students with clinical experience. The scientists participated during an international scientific congress of the European Organisation for Caries Research, whereas the latter two groups were employees or students of the Dental Clinic, Justus Liebig University of Giessen. For all raters, the same set-up was used independent of the setting. They received a short verbal instruction about the procedure, a case report form, and a printout of the photos. No instructions about possible diagnostic criteria were given.

The true status (reference status) of the areas was determined based on a histological evaluation as described in Ganss et al. [10]. The histological evaluation was performed after finishing all assessments in order to ensure perfect blinding. Only 5 of the 49 areas were histologically judged as “dentine not exposed”, resulting in a prevalence of nearly 90%.

The only pre-specified area-related factor given by the domain experts was the tissue loss. Prior to the assessment, tissue loss was quantified according to the criteria of Molnar [13], modified by Ganss et al. [14]. A three-point scale with the levels 1 (minor), 2 (moderate), and 3 (advanced) was used. These criteria do not require any information related to the status of “dentine exposed” or “dentine not exposed”. As pointed out above, tissue loss plays a central role in judgement of the dental wear, and hence the domain experts expected an influence of this factor on the decisions made by the raters. In addition, they expected increasing tissue loss to make the judgement more challenging.

A second pre-specified factor arose from the ordering of the areas, which was identical for all raters. Consequently, if raters performed some type of learning, this should appear in this investigation as an area-specific effect, although it does not, strictly speaking, reflect a property of the area.

### 2.2. Outcome Variables

Areas can vary in difficulty to be diagnosed correctly. Some areas may show clear signs in one or the other direction, whereas other may be highly ambivalent. There may be also area-specific circumstances, which make the judgement in any direction more challenging, e.g., an unusual dark tooth surface. The area-specific rate of correct diagnosis reflects this difficulty, and consequently, this is the first outcome considered. Identifying area-specific factors explaining the variation in this rate may give a hint to what makes an area difficult to diagnose. In the following, we will refer to this area-specific rate, which reflects the number of agreements with the reference status in relation to the histological findings as the “correct rate”.

Areas can vary in their perception by the raters. Some areas may be perceived by most raters as “dentine exposed”, some by most raters as “dentine not exposed”, and some may be perceived in a mixed manner. The rate of positive ratings for an area reflects directly these different degrees of perception; we refer to this as the “positive rate” and it constitutes the second outcome considered. Identifying area-specific factors explaining the variation in the positive rate may give a hint to why raters perceive areas differently.

### 2.3. The Overall Analytical Strategy

The first aim is to describe and explain variation across areas with respect to diagnostic difficulty and perception, respectively. Meta analytical techniques will be used to approach this aim. In addition to the two area-specific factors mentioned in Section 2, a third factor to be considered is the reference status, which is known through histological examination. However, the role of the reference status differs between the two outcomes. For the diagnostic difficulty, the reference status can be handled similarly to the other two factors; areas with “dentine exposed” may be easier to diagnose than areas with “dentine not exposed”, or vice versa, or there may be no difference. An influence of the reference status on the diagnostic difficulty means just that sensitivity and specificity are unequal. For the perception, an influence of the reference status is more or less a prerequisite; without such an influence, sensitivity and specificity would be at chance level. i.e., they would add to 1.0, and there would be no reason to perform this investigation. Consequently, our focus is on variation, which cannot be explained by the reference status.

To identify unmeasured factors, which may explain the variation, two approaches are considered. First, the similarity between areas with respect to the rating is assessed in order to identify pairs of areas rated by most raters in the same manner. Once such pairs similar in rating are identified, they can be investigated with respect to similarities in other aspects. Second, we can make use of the fact that the rating is based on the visual inspection of the areas by the raters, and consequently the input used by the raters can be rather well approximated by the photographs provided. Hence, by presenting the area images together with information on the two outcomes of interest, a visual inspection by the domain experts can be used to identify new, previously unmeasured factors associated with the outcomes of interest.

Once preselected factors or new unmeasured factors are identified, three further questions appear. First, the factors may be correlated and do not represent conceptually independent constructs. Second, it is of interest how much of the variation in the outcomes can be explained by all factors. Third, factors explaining the variation in perception may represent cues used by the raters, and it is of interest to understand whether these cues are used to varying degrees by different raters. Corresponding analyses will also be presented.

### 2.4. Formal Analysis of Variation

Both outcomes considered, the correct rate and the positive rate, are simple relative frequencies varying across areas. This variation can simply reflect sampling variation. Our interest is the variation of the underlying true values, i.e., the area-specific probabilities of a correct diagnosis or a positive rating. These probabilities can be interpreted as the relative frequencies to be seen in an infinite sample of raters. This true variation can be estimated by meta-analytical techniques, using the relative frequencies and their standard errors as input. From such a meta-analysis, estimates of the mean and the standard deviation of the true values can be obtained, as well as a *p*-value testing the null hypothesis of no variation. To facilitate interpretation of the variation, mean and standard deviation are transformed into a 95% range making use of the 1.96σ rule.

Similarly, meta regression can be used to assess the influence of area-specific factors on the true outcome values. R^2^ values are used to describe the contribution of a single factor in explaining the variation. In a first step, these values will be reported based on models with a single covariate. To adjust for the reference status when analysing the positive rate, this variable is added as a second covariate, and the increment in R^2^ is reported. In a final step (reported in Section 3.6), a model with all factors is considered.

### 2.5. Pairwise Similarity of Areas in Rating

The tendency of two areas to be rated in the same manner by many raters can be assessed by the agreement rate of the ratings across the different raters. As the positive rates vary across areas, a simple comparison of agreement rates can be misleading, as the chance level of agreement varies with the positive rate. Consequently, we make use of Cohen’s kappa [15] to assess the similarity. These values were computed for all pairs and the most promising were selected by statistical considerations.

### 2.6. Preparing Feedback on Unmeasured Factors by the Domain Experts

In order to identify potential area-specific factors, which may be associated with the difficulty or the perception, the images of the marked circles to be assessed were arranged in several manners in order to allow a further inspection by the two domain experts. (A) All images were sorted according to the positive rate with additional marking of the five areas with negative reference status. (B) The images from the five pairs with maximal similarity were arranged side-by-side. (C) The images from the same tooth were arranged side-by-side with additional information on the positive rate. The latter allows focusing on within-tooth differences. No specific arrangement according to the difficulty was made, as the sorting would differ only for the five areas marked in (A).

### 2.7. Variation in the Rater-Specific Use of Cues

Once potential area-specific cues used by the raters are identified, the question about a varying or a uniform use of a cue across the raters arises. In the extreme case, some raters may never use this cue and some raters may rely completely on such a cue. However, it is more likely that there is a continuous variation in the sense of a variation in the degree the raters make use of the cue.

This question can be addressed by performing for each rater a logistic regression of the rating vs the factor of interest. The corresponding log odds ratios reflect the degree to which the rater is using the factor as a cue. These log odds ratios are estimates suffering from stochastic variation, but again the meta-analytical approach can be used to estimate the mean and the standard deviation of the true log odds ratios. To facilitate the interpretation of these values, all factors enter these analyses after standardisation to mean 0 and variance 1. These makes the mean values and the standard deviations directly comparable across the factors. Higher mean values imply a stronger association on average, and higher standard deviations a higher variation of the degree of association from rater to rater. In addition to the mean and the standard deviation, estimates of the 95% range of the true log odds ratios and the *p*-value of testing the null hypothesis of no variation across the raters will be reported.

For the order of the areas, it is also possible that the order affects correctness of the rating in the sense of a learning effect. Such a learning effect can again vary from rater to rater. Hence, for the order, we performed the same analysis using the correctness of the rating instead of the rating as outcome.

### 2.8. Overview about the Statistical Methods Used

As meta-analytic technique we made use of a random effects meta-analysis based on the restricted maximum-likelihood estimation principle (REML) as recommended by Langan et al. [16]. This technique was also used to perform meta regressions. The transformation of estimated means and standard deviations into a 95% range using the 1.96-σ rule can lead to boundaries outside of the range (0,1). To avoid this, the latter step is applied after application of a logit transformation to the input values and corresponding back-transformation of the range boundaries.

In computing Cohen’s kappa for all pairs of areas, we will always find some pairs with maximal similarity. To address this issue, we used a parametric bootstrap approach to simulate the expected distribution of the maximal kappa values under chance conditions. This allows us to assess whether we observed larger kappa values than expected under chance conditions, and to determine a choice for the number of pairs to be investigated because of an increased similarity. Details of this approach are given in Appendix B. The pairs themselves were selected as those with the highest values for the lower bound of a 95% confidence interval for Cohen’s kappa, which is a compromise aiming to take both the magnitude of the estimate as well as its imprecision into account.

As we were successful in detecting an unmeasured factor, further statistical methods were applied in a corresponding post-hoc analyses. They are motivated and described in the corresponding part of the Section 3.

All computations were done with STATA (Version 17.0, College Station, TX, USA).

## 3. Results

### 3.1. Variation in Diagnostic Difficulty across Areas and Potential Explaining Factors

Figure 2 depicts the distribution of the correct rates and the association with the tissue loss. A wide range of values from less than 10% to nearly 100% can be observed. In addition, there is a clear association of the diagnostic difficulty with the grading of tissue loss. While the correct rate can be as low as 10% in the case of a minor tissue loss, the lowest rate is over 50% in the case of advanced tissue loss. All five areas in which dentine was not exposed were diagnosed correctly by 80%, which is clearly above the average observed in the other areas. These five observations fit well into the general joint distribution of tissue loss grading and correct rate. On the left side of Figure 3, the association with the order of analysing the areas is depicted, indicating the absence of any association.

The huge variation in diagnostic difficulty could be confirmed in the formal analysis of variation (Column 1 of Table 1 and Appendix A) suggesting true values of the correct rate varying from less than 20% to more than 90%. The association to the three factors is quantified on the left side of Table 2. For both tissue loss as well as the reference status, an association could be established. The latter reflects the difference between sensitivity and specificity already observed in the original analysis of the study. The tissue loss can explain the variation to a higher degree than the reference status.

Estimates of the mean, the standard deviation and the 95% range of the true values of the correct rate or the positive rate, respectively. In addition, the *p*-value referring to the null hypothesis of no variation is reported.

### 3.2. Variation in Perception across Areas and Potential Explaining Factors

Figure 4 depicts the distribution of the positive rate and the association with the tissue loss. Again, a wide range of values from less than 10% to nearly 100% can be observed. However, no association with the tissue loss grading can be observed, as the range is the same in all three categories. However, this result is driven by the five areas with a negative reference status. Stratifying by the reference status, an association can be observed driven by the areas with a positive reference status. The right side of Figure 3 depicts the association with the order of the examination, suggesting the absence of such an association.

The huge variation in the positive rate could be confirmed in the formal analysis of variation (Column 2 to 4 of Table 1 and Appendix A). This holds for both the overall sample as well as within the areas with “dentine exposed”. In the latter, the true values of the positive rate varying from less than 20% to more than 90%. The association to the three factors is quantified in columns 2 and 3 of Table 2. The reference status can explain a third of the variation in the overall sample, and tissue loss can explain 7% of the variation on top of this.

### 3.3. Pairwise Similarity in Rating across Areas

The bootstrap investigations presented in Appendix B suggest that there are about five pairs of areas with a similarity in rating above chance level. Consequently, the five most promising pairs were selected for further inspection. Basic properties of these five pairs are shown in Table 3. Due to the high agreement, the areas within each pair had very similar positive rates. However, across the pairs the positive rate varied substantially. The most similar pair of areas originated from the same tooth and one pair combines an area with a positive reference standard status with an area with a negative reference standard status. Three pairs shared the tissue loss grading, but the other two pairs combined an area with minor tissue loss with an area with advanced tissue loss. The areas 23 and 35 appeared twice among the five most similar pairs and were linked to each other and to the areas 25 and 37, respectively. In addition, the areas 23 and 37 were ranked as the 13th most similar pair. This may hint to a cluster of areas similar in rating.

### 3.4. Search for Unmeasured Area-Specific Factors Related to Diagnostic Difficulty or Perception

The arrangement of the images of the areas analysed are shown in Figure 5, Figure 6 and Figure 7. Figure 5 gives the immediate impression that the positive rate is associated with the background colour of the area to be evaluated. The darker the area, the higher is the positive rate–although there is no one-to-one correspondence. Figure 6 corroborates this impression and indicates that the local disturbance in tooth 23 had no effect on the perception. This is in line with Figure 5, where even more pronounced local disturbances in the areas 6a, 12, and 16 do not seem to affect the perception. Figure 7 illustrates again the role of the background colour: In the pairs 18, 22, and 32 similar background colours correspond with similar positive rates, whereas in the pairs 5 and 26 the darker background colour coincides with the higher positive rate. However, the pairs 14 and 24 remind us that the perception is not a simple function of the background colour: positive rates may differ substantially between two areas with similar background colour from the same tooth.

### 3.5. Post-hoc Analyses of the New Identified Factor

In order to define a numerical variable catching the background colour, the areas within the marked circles were analysed using Adobe Photoshop CS5 Extended Version 12.0 and the colour of each pixel was translated into the decimal RGB colour code, i.e., three coordinates. The lower these codes, the darker is the colour: white is represented by 255, 255, 255, black by 0, 0, 0. To obtain an area-specific value, the median over all pixels was determined. The median was preferred to the mean as the latter is more sensitive to colour disturbances in the image. To investigate whether the positive rate can be explained by these colour values, the positive rate was modelled as a linear combination of the three median values. Instead of ordinary least square regression, median regression was used, which is robust against outliers. Such outliers have to be expected, as in some images the disturbances affect more than half of the pixels, such that even the median values cannot catch the background colour (e.g., area 16 and area 29).

In the fitted model, the regression coefficients for R, G, and B were 0.013, −0.009, and −0.002, suggesting not much influence of the blue component. This coincides with the typically yellow to brown colouring in Figure 5, as the B-coordinate of yellow or brown is 0. The predictions from the fitted model defined a new variable, which is labelled as “colour index” in the sequel.

Figure 8 depicts the relation of this colour index to the positive rate, suggesting some success in catching the relation seen in Figure 5. The figure allows identification of a few areas for which the positive rate is distinctly lower than that predicted by the colour index. Area 16 is an area with a huge dark disturbance such that even the median values reflect the dark disturbance instead of the background colour. Area 13 showed a rather orange, but not necessarily dark tone of the background colour. The colour index seems here to represent more the colour than the darkness. Furthermore, the five areas with a true negative status belong to this group, indicating that the raters could make use of alternative cues to correctly identify these areas, reflected in a low positive rate. The figures also allow identifying a few areas for which the positive rate is distinctly higher than predicted by the colour index. Area 29 is an area with a huge bright disturbance. Area 21 and 26a do not share the brownish-red tone of many of the other areas with a high positive rate.

### 3.6. Independence of Cues and Overall Explained Variation

With respect to the perception represented by the positive rate, three factors with an association could be identified so far: the tissue loss, the colour index, and the reference status. To understand their overall role, the associations among these factors are of interest. The tissue loss grading and the colour index show a Spearman correlation of 0.06, and hence represent two independent cues probably used by the raters. The mean values of the colour index are very similar in areas with “dentine exposed” (0.66) and in areas with “dentine not exposed” (0.63), as also visible in Figure 8. Hence, there is no association between the colour index and the reference status. The tissue loss is somewhat associated with the reference status, as areas with a minor tissue loss are never “dentine not exposed” (see Figure 2 and Figure 4). However, results from a multiple regression analysis suggest that all three factors contribute to explain the variation in the positive rate (Table 4). Overall, the three factors could explain 64.2% of the variation.

### 3.7. Variation in the Rater-Specific Association

Table 5 summarises the results from analysing the distribution of the rater specific log odds ratios expressing the association between area-specific factors and the rating (or correctness of the rating). The strongest association on average as well as the highest variation across raters could be observed for the colour index. The 95% range suggests that the rating of all raters showed some association with the colour index, but to a highly varying degree. After adjustment for the reference status, the tissue loss grading showed an association on average and a variation across raters in the magnitude of 50–60% of the values observed for the colour index. The 95% range indicates again that the ratings of all raters show an association with the tissue loss grading, but to a highly varying degree.

Without adjustment, no variation could be observed in the association with the tissue loss. The same holds for the association between the order and the correct diagnosis. Positive ratings may have slightly decreased over time with a slight variation across raters. However, for all five analyses, the variation across the raters was not statistically significant.

## 4. Discussion

### 4.1. Background

Diagnosis is a multifaceted operation. One way to describe it is to divide the decision-making process into two systems, associative and rule-based [17]. Both influence each other, and can have different dominance in diagnostic situations. Dual-process theory describes both as part of a model that provides a theory for understanding decision-making in medicine and can explain the performance of physicians or, in our context, of raters [18]. Based on this model, factors that influence (diagnostic) decisions in medicine and can lead to diagnostic inaccuracies have been discussed [5].

The fact that examiners differ in their diagnoses is a common phenomenon (in our context, for example [19]), which leads to the need for calibration and training (for example [20,21,22]). Depending on the difficulty and complexity of the diagnostic question, the agreement of raters can vary even after such calibration and training sessions; for example, the recording of plaque scores on images seems to be possible with very good agreement [20], while the assessment of study casts with regard to tooth wear does not seem to work satisfactorily [21]. Especially when many examiners are involved, it also becomes evident that some raters show less agreement even after calibration sessions than others, and/or need longer training.

Although this issue is well known on the one hand, and on the other hand, the corresponding potential influencing factors on raters decision making have been described, to the best of the authors’ knowledge, the two have not been considered together in depth; therefore, our aim was not only to consider the inter-rater variation or the accuracy of the diagnosis in relation to the reference but to relate it to object- and rater-specific parameters in order to gain more insight into the diagnostic process.

For such a purpose, data that have been collected from as many raters as possible, that are based on as many diagnoses as possible in a standardised setting, and that also include a “true diagnosis” in the sense of a gold standard or reference seem particularly suitable. Furthermore, the diagnoses should not be straightforward and unambiguous, but rather should be those in which, due to their complexity, potential influencing factors can become effective. The data from dentistry used here seemed to us to be well suited for our research question.

### 4.2. Analytical Approaches

Understanding variation across the objects to be diagnosed by inspection of object-specific rates of correct or positive ratings is a non-trivial task, as they are also subject to random variation. Meta-analytical techniques allow to separate true variation from random variation and to assess the influence of object-specific factors in an adequate manner. Fortunately, such techniques are today available in standard statistical packages, and this makes the analyses presented in this paper feasible.

Looking for associations with unmeasured factors is challenging. As in this setting, factors associated with the visual impression could be expected, a visual inspection after a corresponding rearrangement was a rather straightforward approach. Studying the agreement in ratings for pairs of objects or comparison within predefined object groups are techniques which can be used in general. Post-hoc analyses of the association with an unmeasured factor identified can be based on various techniques, but inferential analyses should be interpreted with caution, as the hypotheses considered may be data driven.

### 4.3. Results with Respect to Predefined Factors

As mentioned above, the criterion “dentine exposed or not” is part of most clinical evaluation systems for quantifying dental hard tissue loss. Although these indices have been used for decades, the question of the extent to which this distinction can be made in a reliable manner in a clinical context at all has not come into view for a long time. The publication [10] from which the present data are taken has attempted to approach this question. The results of the study had shown that the diagnosis of exposed dentin was difficult and that there was considerable variability in diagnostic decisions. Both seemed thus well suited for analysing further explanatory factors in this context. In this study, those teeth were selected from a pool of extracted teeth that showed minor, moderate, or pronounced tissue loss in roughly equal proportions and in which a balanced number of both diagnoses could be expected accordingly. The fact that the histological assessment carried out at the end of the examination phase revealed almost exclusively exposed dentine was entirely unexpected. On the one hand, this leads to limitations of the present analysis. As the correct rate and the positive rate differ only in five teeth, the question whether a factor is related to diagnostic difficulty or to perception cannot be well separated. For the tissue loss, we could establish a relationship to both, which is mainly driven by the association within the areas with “dentine exposed”. On the other hand, it also gives rise to interesting aspects.

Expectation and visual priors may influence perceptions [23]. In the present context, expected prevalence could be one factor [24]; thus one could suppose that the raters would assume a priori roughly the same number of cases with exposed and non-exposed dentine. While at the beginning they might have looked specifically at the doubtful cases without prejudice, they might have realised later that they had already made the diagnosis “exposed dentin” very frequently, and may have adapted their default for doubtful cases. However, we could not find evidence for timing effects on the positive rate in this study. This is similar to findings in another context: a scoping review on factors which might influence performance in detecting lung cancer [24] found that the expected prevalence of malignant lung nodules did not affect the sensitivity of diagnoses. However, the raters took more time to diagnose when they expected a high prevalence. Whether this was also the case in the present study could not be determined.

One pre-specified area-specific factor we could investigate was the severity of the tissue loss of the dental crown. Taking into account the widely accepted role of the severity of tissue loss in judging dental wear, the domain experts expected that at least some raters would regard this as a cue to judge whether dentine is exposed. This could be corroborated by our analyses as the positive rate increased with increasing severity of the tissue loss, if it is taken into account that the reference status has the expected strong impact on the probability of a positive rating. Moreover, there is some evidence that this association is present in all raters, but to a highly varying degree.

There was a substantial variation in diagnostic difficulty across the areas, and this allowed us to confirm the tissue loss as an area-specific factor influencing the diagnostic difficulty. A minor tissue loss makes the diagnosis more challenging. This suggests using the tissue loss as a stratifying factor in judgement training and diagnostic accuracy studies with the ultimate goal to achieve high accuracy also in the presence of minor tissue loss.

### 4.4. Results with Respect to Unmeasured Factors

In addition to investigating known and pre-specified factors that may explain the variation in raters, the detection of unmeasured factors influencing the difficulty of diagnosis or perception may be also highly relevant. For this purpose, it may be helpful to sort the objects according to the observed rates of correct or positive rating. This can help to identify characteristics that are associated with diagnostic difficulty or perception.

Accordingly, in the present study, the areas examined were sorted in descending order of positive rate. This showed that areas with a darker colour were obviously judged as “dentine exposed” with significantly less variability than those with a lighter colour. This first impression led to a subsequent analysis, which indicated that the background colour of the area being assessed was apparently also a factor influencing the decisions made by the raters.

The question then arises as whether such a newly identified factor is in fact logically linked to the condition to be diagnosed and hence represents a useful cue, or whether the association reflects the use of a useless cue, based on an assumed but not existing association, or whether the association reflects a spurious association to another cue actively used. It is generally believed that as enamel thickness decreases, the underlying dentine makes the tooth appear more yellowish. Therefore, it has been suggested that tooth colour should be used as an indicator of erosion progression [25]. Although this relationship could be confirmed in principle, interindividual differences were so large that tooth colour does not appear to be suitable as an indicator of remaining enamel thickness [25]. A recent study confirmed this finding, i.e., a weak correlation between enamel thickness and tooth colour [26]. Hence, these experiments indicate that dentine can be exposed with very different tooth colours and that lighter tooth colour is not a reliable indicator of intact enamel coverage. However, the raters seem to have succumbed to the prejudice that a darker tooth colour is indicative of exposed dentin. Again, there is some evidence that the raters did this to a varying degree. This reflects perhaps the uncertainty about the value of this cue. If the raters are to a certain degree guided by the colour of the teeth, this could also have further implications: The colours of the surroundings of the teeth to be judged, for example the skin or lip colour (but perhaps also the gum colour) could influence the perception of the tooth colour [27,28] and thus represent a potential cognitive bias.

### 4.5. Implications for Diagnosing in Dentistry

The variation in making use of cues across raters and the higher influence of the colour of the area examined on the raters’ decisions than the tissue-loss grading indicates that traditional ideas and general expectations can represent a considerable cognitive bias in decision-making about the exposure of the dentine. This suggests the need for a broader discussion about the favourable and unfavourable cues to be used in diagnosing the exposure status of the dentine. Preferably, such a discussion should be informed by dedicated investigations of the value of different potential cues. A practical conclusion from this further analysis of the data on the diagnosis of exposed dentin could be that raters should be specifically trained not to take into account tooth colour and the extent of substance loss, possibly seeking new criteria that are better suited for correct diagnosis but also for better agreement between raters; for example, the fine contour of the tooth surface and its light-reflecting properties, or perhaps even the realisation that the diagnostic question cannot be answered with sufficient accuracy, using only clinical visual diagnosis, and that other diagnostic techniques need to be developed.

The present results indicate that the visual diagnosis of whether dentin is exposed or not is difficult. However, there is no other clinical procedure that can reliably enable such a diagnosis. Therefore, the assessment of the individual case’s risk of progression should always be supplemented by additional information such as dietary habits, diseases that can lead to intrinsic acid exposure of the teeth, e.g., reflux diseases, and physical factors such as bruxism [29]. Whether individual wear is truly progressive can be confirmed for instance by consecutive study casts [30] or with intraoral scanners which are also a suitable tool for quantifying tissue loss [31]. For example, a study on the progression of occlusal wear showed that cuppings on cusps, where it is assumed that dentin is more likely to be exposed, progress faster than other forms of wear [32]. Only all these factors taken together can form a valid basis for a decision on treatment.

The insight that examiners include certain associative assumptions in their decisions to varying degrees should, however, also be taken into account in another context. Recently, more and more automated diagnostic procedures based on artificial intelligence and machine learning are being developed [33]. Ultimately, however, these algorithms are trained by human raters and, in the next step, their output is evaluated by human raters. However, often only a few raters are included for training (e.g., [34]) as well as for evaluation [33], and how these raters are calibrated and trained, and how validly these raters can diagnose, is usually not addressed. Future studies should therefore investigate the influence of rater bias and training and evaluation methods on the performance of algorithm-based methods.

## 5. Conclusions

Our investigation demonstrated that the use of multiple raters in a diagnostic accuracy study allows performing detailed investigations with respect to the variation in diagnostic difficulty and perception. These investigations allow us to obtain further insights into the decision-making process of the raters. This can help to identify favourable or unfavourable cues currently used in diagnosis and conditions challenging the correct diagnosis. Such insights can be used to design training programs and future studies, aiming at better agreement values among raters, better comparability of studies, and better diagnostic certainty in individual patients.

The use of multiple raters requires access to a relevant rater population and the use of non-expensive diagnostic procedures. Many diagnostic procedures in dentistry are based on visual or manual investigations, and, hence, fulfil the latter criteria. Huge clinical departments, collaborations with practicing dentists and frequent scientific, specialised conferences allow recruiting relevant rater populations. Hence, dentistry can be seen as a perfect field for this type of studies.

## Figures and Tables

**Figure 1 ijerph-20-01781-f001:**
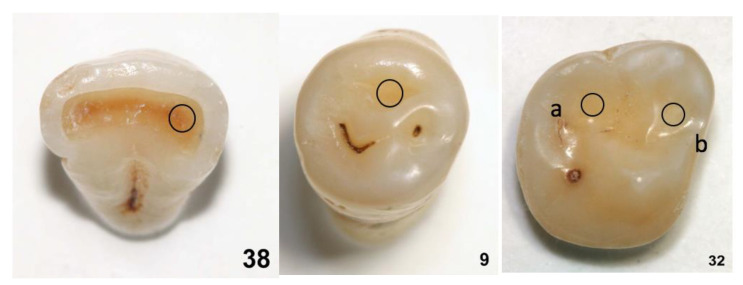
Representative selection of teeth (one incisor, one premolar and one molar) from the total set of teeth examined; the circles indicate the areas that were to be assessed. In the teeth with the numbers 38 and 9, dentin was exposed in the labelled areas, whereas in the tooth with the number 32 it was neither in area a nor in area b.

**Figure 2 ijerph-20-01781-f002:**
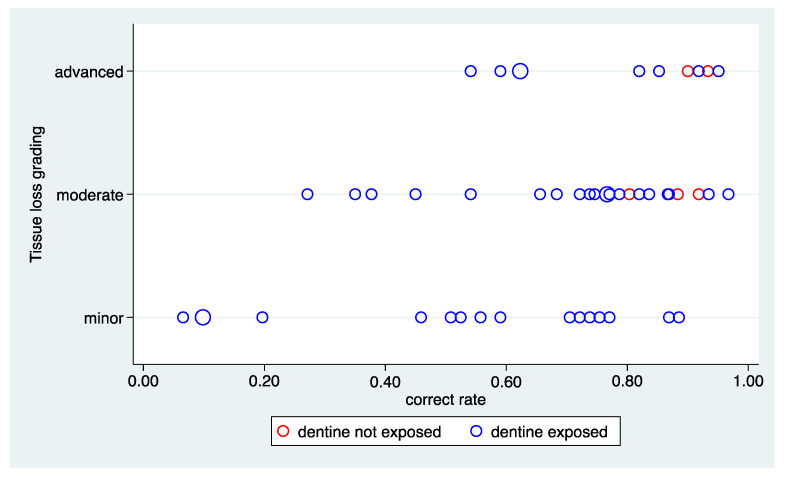
Bubble plots of the correct rate and grading of tissue loss. The correct rate was stratified by the reference status. The area of the bubbles is proportional to the number of observations.

**Figure 3 ijerph-20-01781-f003:**
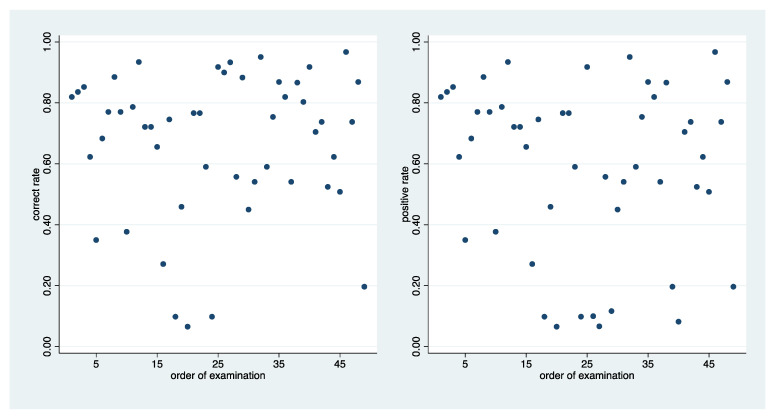
Scatterplots of the correct rate (**left** side) and the positive rate (**right** side) vs. the order of analysing the areas.

**Figure 4 ijerph-20-01781-f004:**
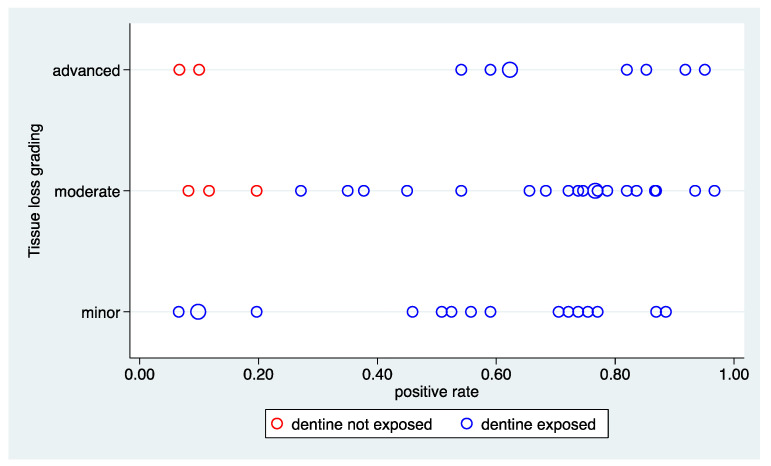
Bubble plots of the positive rate stratified by the grading of tissue loss. The area of the bubbles is proportional to the number of observations.

**Figure 5 ijerph-20-01781-f005:**
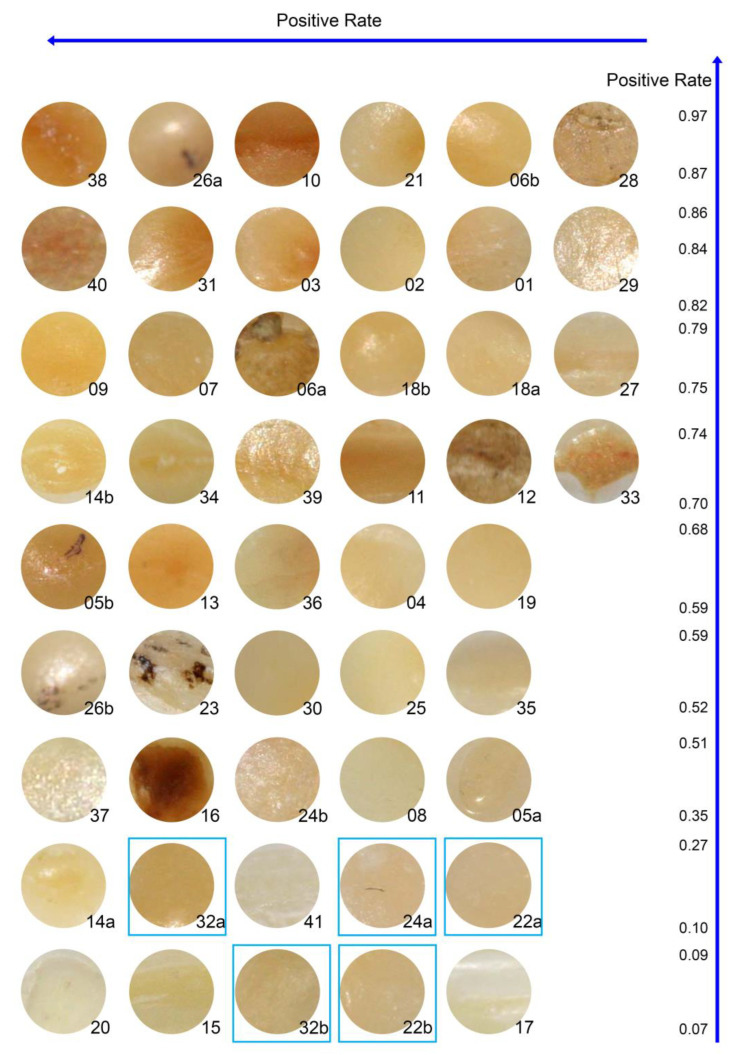
Images of the areas to be. Images are sorted by positive rate (right scale) descending from top to bottom and from left to right within each row. Blue frames indicate areas with “dentine not exposed”. Images are labelled with the area indicator.

**Figure 6 ijerph-20-01781-f006:**
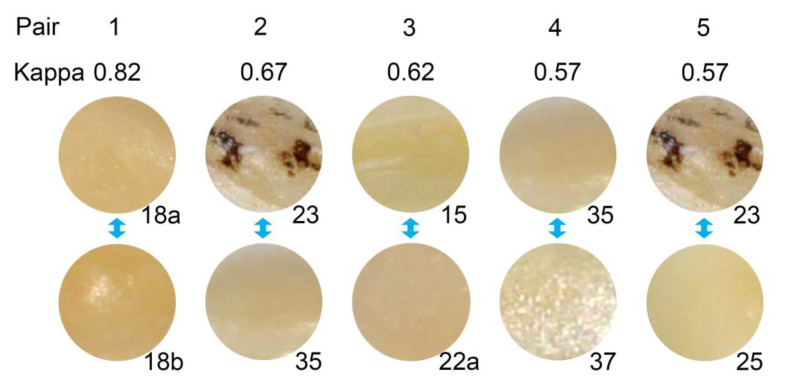
Images of the 5 area pairs with highest kappa values. Images are labelled with the area indicator. The observed kappa values are shown for each pair at the top of the figure.

**Figure 7 ijerph-20-01781-f007:**
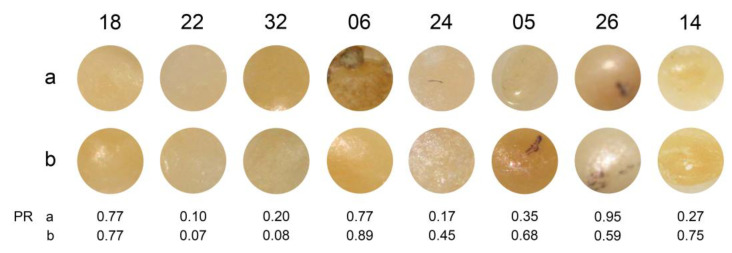
Pairs of images of areas from the same tooth sorted by increasing differences of positive rates. Images are labelled with the area indicator. The observed positive rates are shown at the bottom of the figure.

**Figure 8 ijerph-20-01781-f008:**
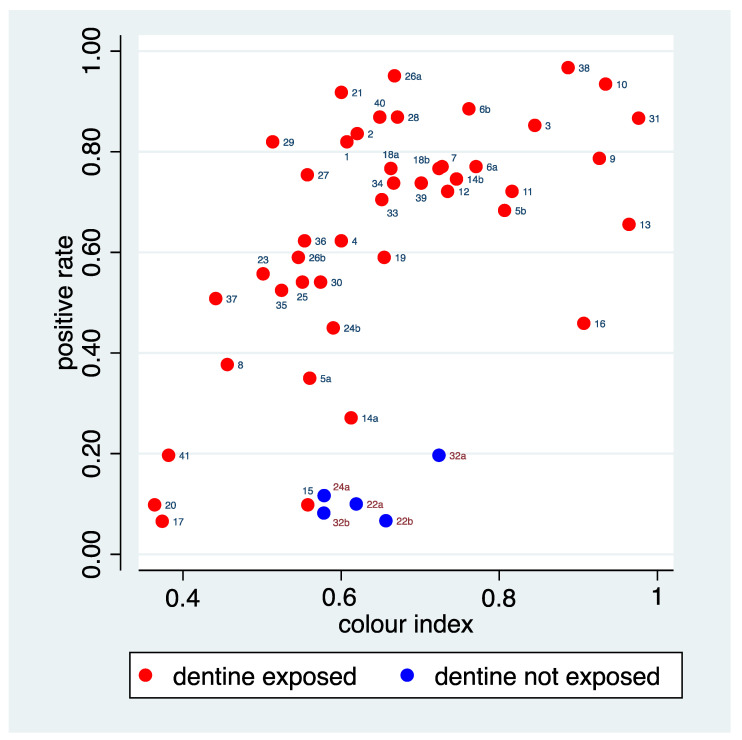
The joint distribution of the colour index and the positive rate. The points are labelled with the tooth number assigned.

**Table 1 ijerph-20-01781-t001:** Estimates of the true values of the correct rate or the positive rate.

	Correct Rate	Positive Rate	Positive Rate within “Dentine Exposed”	Positive Rate within “Dentine Not Exposed”
Mean	0.67	0.59	0.64	0.10
SD	0.23	0.28	0.23	0.004
95% range	0.18–0.96	0.09–0.96	0.17–0.95	0.07–0.19
*p*-value	<0.001	<0.001	<0.001	0.213

**Table 2 ijerph-20-01781-t002:** Amount of variation explained by the single factors depicted by R^2^ values. In addition, the *p*-value referring to the null hypothesis of no influence of the factor on the outcome is given. In the analysis adjusted for reference status, R^2^ values refer to the increment compared with a model with reference status as the only covariate.

	Correct Rate		Positive Rate		Positive Rate (Adjusted for Reference Status)	
	Explained Variation	*p*-Value	Explained Variation	*p*-Value	Explained Variation	*p*-Value
Tissue loss grading	15.0%	0.003	0.0%	0.435	6.5%	0.017
Ordering of areas	0%	0.791	0.0%	0.457	0%	0.957
Reference status	8.0%	0.024	33.9%	<0.001	-	-

**Table 3 ijerph-20-01781-t003:** The five area pairs with highest agreement (kappa value) in the rating. For each pair the positive rate, the agreement rate (a), and the kappa value (κ) are reported. In addition, the reference standard status and the grading of tissue loss are shown.

Pair	Area	Positive Rate	Statistics	Reference Status (Histology)	Tissue Loss Grading
1	18a	0.75	a = 0.93	exposed	moderate
	18b	0.75	κ = 0.82	exposed	moderate
2	23	0.56	a = 0.84	exposed	minor
	35	0.52	κ = 0.67	exposed	minor
3	15	0.10	a = 0.93	exposed	minor
	22a	0.10	κ = 0.63	not exposed	advanced
4	35	0.52	a = 0.79	exposed	minor
	37	0.51	κ = 0.57	exposed	minor
5	23	0.56	a = 0.79	exposed	minor
	25	0.54	κ = 0.57	exposed	advanced

**Table 4 ijerph-20-01781-t004:** Additional variation in positive rate explained by the single factors. Amount of additional variation explained by the single factors depicted by incremental R^2^ values in comparison of a model with only the two other factors. In addition, the *p*-value referring to the null hypothesis of no influence of the factor on the outcome in a model with all three factors is given.

	Positive Rate	
	Additional Explained Variation	*p*-Value
Colour index	23.7%	<0.001
Tissue loss grading	4.5%	0.017
Reference status	37.8%	<0.001

**Table 5 ijerph-20-01781-t005:** Analysis of the distribution of the true values of the rater-specific log odds ratios. Estimates for the mean, the standard deviation, and the 95% range are given. The *p*-value refers to a formal test of the null hypothesis of no variation across the raters.

	Mean	SD	95%-Range	*p*-Value
Colour index	0.752	0.181	0.396–1.107	0.119
Tissue-loss grading	0.145	0.000		0.749
Tissue-loss grading adjusted	0.392	0.104	0.189–0.596	0.317
Order-rating	−0.132	0.061	−0.25–−0.013	0.400
Order-correct diagnosis	0.035	0.000		0.517

## Data Availability

The dataset used is available from the corresponding author on reasonable request.

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
