# Peer review of "Understanding the Influence of Patient Factors on Accuracy and Decision-Making in a Diagnostic Accuracy Study with Multiple Raters—A Case Study from Dentistry"

_ijerph, 2023, doi:10.3390/ijerph20031781_

Round 1

Reviewer 1 Report

As the paper is intended for practical implications for improving diagnostic criteria in the dental community, it is suggested to present conclusions from a clinical perspective in addition to the suggestion of implementing the mentioned statistical techniques. 

Reviewer 2 Report

Abstract: - Please delet the subtitles   Introduction: - The intro is so poor and need references, especially form line 39-66 - Please clarify the originality of the present study   Methods: - Figure 1: please correct the number of each image, why 38, 9 and 32? - Please more informations about the statistical tests   Results: - Please more explanations Figures 2,3 and 4 - Where is Figure 6? - Figure 5 and Figure 7: more details in the legend should be presented   Discussion: Very poor and comparison with other similar studies should be provided, Only two references are presented !

Reviewer 3 Report

1. The abstract is properly formatted and sufficiently clear.

2. The introduction is short. In total, only 11 articles are cited in the article, which is extremely few. In the introduction itself, from lines 38 to 66, no cited article is mentioned, but the authors' opinions are described. This part should be included in the discussion, not in the introduction. Please add more cited articles and at the end of each paragraph mention the author of the information!

3. The materials and methodology are described very well!

4. The results are explained and illustrated clearly and comprehensibly enough for the readers!

5. The discussion part is very weak. Only two authors are mentioned! It should be enriched! The purpose of the discussion section is to compare the authors' results with similar studies by other authors. Please correct!

6. The conclusion is clear and short!

Reviewer 4 Report

Dear authors, your manuscript is very interesting. Please add a statement at the end of the discussion that a medical history is important in addition to the visual assessment in order to determine the cause of dentin exposure. In addition, it is important to investigate occlusal relationships, other local factors that may contribute to tissue loss. It is also extremely important to observe the lost tissues over time in a given patient. Photographs taken of the patient and cast models can be helpful here to determine the progression of the lesion over time. In general, please add that visual assessment alone is insufficient to make a diagnosis and to choose the right treatment.

Round 2

Reviewer 2 Report

Figures 5-7 need more resolution

Problem in the appearance of references in the discussion

Author Response

Dear reviewer, thank you for the time and effort you have put into reviewing the manuscript again.

In order to improve the quality of the figures, we have edited the original photos again and saved them with a higher resolution. We hope that we have also been able to improve the visual presentation.

Regarding the references: we have checked everything and hope that the problems have now been solved. If contrary to our expectations, problems still occur, we would be pleased to receive more details so that we can correct them in a targeted manner.

Reviewer 3 Report

The article is very good! 

Author Response

Dear reviewer, thank you for the time and effort you have put into reviewing the manuscript again. And many thanks for the warm words.